# Enhanced Senescence Process is the Major Factor Stopping Spike Differentiation of Wheat Mutant *ptsd1*

**DOI:** 10.3390/ijms20184642

**Published:** 2019-09-19

**Authors:** Zhixin Jiao, Junchang Li, Yongjing Ni, Yumei Jiang, Yulong Sun, Junhang An, Huijuan Li, Jing Zhang, Xin Hu, Qiaoyun Li, Jishan Niu

**Affiliations:** 1National Centre of Engineering and Technological Research for Wheat/Key Laboratory of Physiological Ecology and Genetic Improvement of Food Crops in Henan Province, Henan Agricultural University, Zhengzhou 450046, Henan, China; zxjiao2018@163.com (Z.J.); chang_top@163.com (J.L.); nxyjym@henau.edu.cn (Y.J.); ylsun1994@163.com (Y.S.); jhan68@163.com (J.A.); lhj19960901@163.com (H.L.); jzhang1023@126.com (J.Z.); liqiaoyun@henau.edu.cn (Q.L.); 2Shangqiu Academy of Agricultural and Forestry Sciences, Shangqiu 476000, Henan, China; nyj317@163.com (Y.N.); huxin2699552@163.com (X.H.)

**Keywords:** wheat (*Triticum aestivum* L.), mutant *ptsd1*, spike differentiation, *senescence*-*associated genes* (*SAGs*), homeotic gene, phytohormone, Ca^2+^ signaling

## Abstract

Complete differentiation of the spikes guarantees the final wheat (*Triticum aestivum* L.) grain yield. A unique wheat mutant that prematurely terminated spike differentiation (*ptsd1*) was obtained from cultivar Guomai 301 treated with ethyl methane sulfonate (EMS). The molecular mechanism study on *ptsd1* showed that the *senescence-associated genes* (*SAGs*) were highly expressed, and spike differentiation related homeotic genes were depressed. Cytokinin signal transduction was weakened and ethylene signal transduction was enhanced. The enhanced expression of Ca^2+^ signal transduction related genes and the accumulation of reactive oxygen species (ROS) caused the upper spikelet cell death. Many genes in the WRKY, NAC and ethylene response factor (ERF) transcription factor (TF) families were highly expressed. Senescence related metabolisms, including macromolecule degradation, nutrient recycling, as well as anthocyanin and lignin biosynthesis, were activated. A conserved tae-miR164 and a novel-miR49 and their target genes were extensively involved in the senescence related biological processes in *ptsd1*. Overall, the abnormal phytohormone homeostasis, enhanced Ca^2+^ signaling and activated senescence related metabolisms led to the spikelet primordia absent their typical meristem characteristics, and ultimately resulted in the phenotype of *ptsd1*.

## 1. Introduction

Wheat spike differentiation is a key developmental stage as a transition from vegetative growth to reproductive growth in wheat (*Triticum aestivum* L.) [1]. Complete differentiation of the spikes guarantees the final wheat grain yield [2]. In general, the indeterminate inflorescence meristem of wheat can initiate conversion into spikelet meristem, followed by the initiation of glume primordia, lemma primordia and ultimately form a unique reproductive inflorescence unit termed a spikelet [1]. The wheat spike mutants are ideal germplasm resources for molecular genetic studies on wheat spike development. Up to now, only one “degenerated spike” mutant is obtained from EMS treated winter wheat *Jing411* [3]. The upper spikelets develop slowly and cease growth completely after heading, however, the molecular mechanisms remain unknown. Understanding the molecular mechanisms of spike degeneration will be valuable for improving wheat yield potential.

The ABCDE model for plant flower development proposes that floral organ identity is defined by five classes of homeotic genes, named A, B, C, D, and E [4]. Generally, the ABCDE model is equally fit for monocots like rice and wheat [5,6]. According to this model, the class A genes determine lemma and palea identity, the class B genes specify lodicule and stamen identity, the class C and D genes specify stamen and ovule identity, and the class E genes specify organ identity and determine the spikelet meristem [6,7]. Wheat *TaFL* (*FLORICAULA/LEAFY* ortholog) is associated with spikelet formation [8]. Wheat Flowering locus T2 (*TaFT2*) is expressed in the distal part of the developing spikes and contributes to the regulation of the number of spikelets per spike [9]. The miRNA172-*Q* gene (*AP2* like gene) system plays a crucial role in spike morphogenesis [10]. Overexpression of miRNA172 leads to elongated spikes [11]. Although a few genes related to spike differentiation have been reported in wheat, the gene regulatory network remains largely unknown.

Plant hormones are key regulators of spikelet differentiation and degeneration [12]. In rice, the auxin signaling is related to the phenotype of premature termination of spikelet development in *aberrant spikelets and panicle1* (*asp1*) mutant [13]. Cytokinins (CKs) can enhance plant growth and promote floret development [12]. In contrast to auxin and CKs, ethylene (ET) is generally regarded as an inhibitory growth regulator [14]. ET plays a vital role in floret degeneration and spikelet degeneration in the plant [15,16]. However, the molecular mechanism of the cross-talk among hormones in suppressing or enhancing spikelet degeneration remains unknown.

Senescence affects vegetative and productive developmental processes in plants, several *SAGs* have been discovered in model plant *Arabidopsis thaliana* [17]. During wheat senescence processes, proteins are degraded, and nutrients are re-mobilized from senescing leaves to other organs [18]. Many genes involved in metabolism, degradation and regulator processes have been identified from a wheat premature leaf senescence mutant *m68* [19]. A number of leaf senescence regulatory genes are identified, notably NAC-domain and WRKY genes [20,21]. A NAC gene (*Gpc-B1*) is isolated from wild emmer wheat (*Triticum turgidum* ssp. *dicoccoides*), and it can accelerate senescence and increase nutrient remobilization from leaves to developing grains [22]. There are some reports about wheat senescence of vegetative organs, however, little is known about that of productive organs [18,19,23]. Some miRNAs participate in regulating leaf senescence such as miR164. The feed-forward signaling cascade involving *ETHYLENE-INSENSITIVE2* (*EIN2*), *EIN3, miR164*, and *ORE1* plays an important role in the processes of leaf senescence in *Arabidopsis* [24,25].

Previously, we have obtained several stable hereditary prematurely terminated spike differentiation (*ptsd*) mutants from the EMS treated wheat cultivar “Guomai 301” and “Shengnong 1” [26,27]. Among them *ptsd1* is one of the carefully researched mutants. Most plants of *ptsd1* only grow several grains, while the extreme individuals grow no grains completely. Genetic analysis indicates that it’s regulated by recessive genes. Up to now, we haven’t found such kind of mutants have been reported in wheat. It provides an opportunity for us to explore how wheat regulates its spike differentiation. To analyze the molecular network regulating the abnormal spike differentiation of mutant *ptsd1*, we compared the transcriptomes and miRNomes of the wide type (WT) and mutant *ptsd1* at the early spike differentiation stage. Our results suggested that *SAGs* were highly expressed, programmed cell death (PCD) was enhanced, ethylene signal transduction related genes and the degradation related genes of cytokinins were highly expressed in *ptsd1*. Global analysis of the metabolism related differentially expressed genes (DEGs) and measurement of the metabolism indicators revealed that the degradation of macromolecules and the biosynthesis pathway of phenylpropanoids were enhanced in mutant *ptsd1*.

## 2. Results

### 2.1. The Upper Spikelets of the Mutant ptsd1 Didn’t Differentiate

The agronomic traits of *ptsd1* and its wild type parent (WT) were compared, especially their spikes were carefully observed during differentiation and development. The average spike length, fertile spikelet number and seed number per spike of the *ptsd1* were only 4.19 cm, 4.50 cm and 14.16 cm respectively, they decreased by 60.0%, 77.5% and 80.0% compared to those of the WT (Figure 1A,B, Appendix A).

In WT, each spike had 19–21 fertile spikelets (average number was 20) (Appendix A), each spikelet primordium differentiated into a spikelet comprising 3–5 florets. The first visible differentiated structures were glume primordium and lemma primordium, and then one floret primordium, the three stamen primordia and the pistil primordium in each floret can be seen as independent structures at the terminal spikelet stage (Figure 1C). In *ptsd1*, each spike only had 3–6 fertile spikelets (average number was 4.5) (Appendix A). The differentiation of the upper spikelets terminated prematurely at about late double ridge stage (Appendix A), the spikelet primordia only formed some small bumps in the end (Figure 1D), however, the under spikelets differentiated almost normally (Figure 1D). The date of apex differentiation (transition period) of *ptsd1* was about one week later than that of WT, and the dates of heading, anthesis, and maturity were about ten days later than those of the WT (Appendix A).

### 2.2. The Meristem Characteristics of the Undifferentiated Spikelet Primordia on the ptsd1 Were Absent

The enlargement of spikelet primordia proceeded in both acropetal and basipetal orientations during spike morphogenesis at late double ridge stage [28] in WT (Appendix A). The spikelet primordia possessed typical meristem characteristics, such as small cells, none or very small vacuoles, dense cytoplasm, relatively bigger nuclei and thinner cell wall (Appendix A). However, the upper spikelet primordia of *ptsd1* didn’t differentiate, and the bottom spikelet primordia initiated differentiation at this stage (Appendix A). At the initiation of the second awn stage, the stamen and pistil primordia could be seen in both WT and the bottom spikelets of *ptsd1* (Appendix A), however, the upper spikelet primordia of *ptsd1* were still at about the double ridge stage without typical meristem (Appendix A).

### 2.3. Gene Transcripts Were Abundant in ptsd1

To eliminate the interference of genes involved in stem development and identify spike differentiation specific genes, the “stem tips” below young spikes were used as control samples. A total of 131.73 G data were obtained from the four super bulked samples at late double ridge stage: young stem tips of WT (WT-ST; T1a, T1b, T1c), young spikes of WT (WT-YS; T2a, T2b, T2c), young stem tips of *ptsd1* (*ptsd1*-ST; T3a, T3b, T3c) and young spikes of *ptsd1* (*ptsd1*-YS; T4a, T4b, T4c). The mapping rates of the reads in each library to reference wheat genome ranged from 79.23% to 82.46% (Appendix A). The Pearson’s correlation coefficients among the three replicates for each sample ranged from 0.9323 to 0.9946 (Appendix A), indicating that the three biological replicates were consistent. The principal components analysis indicated that the three biological replicates were clustered together in one ellipse (Appendix A).

A total of 4439 genes were significantly differentially expressed between *ptsd1*-YS and WT-YS. Among them 3532 DEGs were highly, only 907 DEGs were lowly expressed in *ptsd1*-YS (Appendix A) (A/B, Up: the expression level in A was higher than B, and down: the opposite.). Obviously, the expressions of most DEGs in differentiating spikes of *ptsd1* were higher.

### 2.4. Senescence Related Biological Process in Mutant ptsd1

A total of 4871 genes were significantly differentially expressed among WT-YS/WT-ST, *ptsd1*-ST/WT-ST, *ptsd1*-YS/WT-YS and *ptsd1*-YS/*ptsd1*-ST (Appendix A). All the DEGs were clustered by *k*-means setting the *k* as seven (Appendix A). These clusters displayed various gene expression patterns which were correlated with their functions. The abnormal expression patterns of the DEGs led to the prematurely terminated spike differentiation in *ptsd1*.

There were 364 and 204 DEGs in clusters 1 and 5, and they were most highly expressed in WT-YS and lowly expressed in *ptsd1*-YS (Figure 2A). Enrichment analysis of these genes in clusters 1 and 5 revealed that most of them were involved in spike differentiation related biological processes such as the top three GO terms of “flower development, meristem development”, and “cell differentiation” (Figure 2C). This indicated that the spike differentiation related genes were lowly expressed in *ptsd1*-YS. Contrarily, 340 and 2004 genes in clusters 2 and 7 were most highly expressed in *ptsd1*-YS (Figure 2A). Enrichment analysis indicated that most of them were involved in senescence related biological processes such as the top five GO terms of “cell wall macromolecule catabolic process” which involved in cell wall macromolecule degradation, “glutathione transferase activity” involved in detoxification, “chitin binding” involved in chitin degradation, “systemic acquired resistance, salicylic acid mediated signaling pathway” involved in PCD and the “negative regulation of PCD” involved in cell senescence (Figure 2C). This indicated that senescence related genes were highly expressed in *ptsd1*-YS. The qRT-PCR analyses showed that the expression patterns of eight representative genes in classes 1, 2 and 7 were well consistent with that of the sequencing results (Figure 2B). The spike differentiation related genes (*TaSEP5*, *TaSEP6* and *TaFL*) in cluster 1 were lowly expressed in *ptsd1*-YS, the senescence related genes (*TaCht*, *TaGSTF2* and *TaGS1a*) in cluster 7 were highly expressed in *ptsd1*-YS.

### 2.5. Functional Categorization of the DEGs Between ptsd1-YS and WT-YS

The genes were considered as not expressed if their average fragments per kilobase of transcript per million fragments mapped (FPKM) < 1 in samples. Ultimately, 2582 DEGs had at least fourfold change in transcript abundance between *ptsd1*-YS and WT-YS were identified (Appendix A), and they were sorted into 10 putative functional groups according to their annotation (Figure 3). The unclassified (325) and unknown function (659) DEGs took up 38% of the total DEGs (Appendix A). The other functional categories of the genes in the central pie chart comprised regulation (25.3%), protein and amino acid metabolism (9.8%), stress/defence (6.4%), transport (6.0%), secondary metabolism (6.2%), lipid metabolism (3.9%), carbohydrate metabolism (3.6%) and nucleotide metabolism (0.7%) (Appendix A). The genes encoding putative regulatory proteins in the left pie chart and the genes involved in macromolecule degradation of four major nutrients in the right pie chart are discussed in detail in the following sections (Figure 3).

### 2.6. Hormone and TF-Related Genes Were Highly Expressed in ptsd1

A total of 132 DEGs (FC ≥ 2) were associated with various aspects of hormone homeostasis, such as biosynthesis, response, signaling and metabolism (Appendix A). The top two groups of the significant DEGs were related to ET and CK (Appendix A). Among ET signal transduction-related genes, twelve homologs of ethylene insensitive (EIN) genes were highly expressed in *ptsd1*. Meanwhile, several ET signal-related genes were highly expressed in *ptsd1*, including the homologs of ethylene response factor (Appendix A). The enhanced expression of ET-biosynthesis and signal transduction related genes in *ptsd1* suggested that more active ET signaling might be a major factor leading to the phenotype of *ptsd1*. A total of 25 CK metabolism and response genes were differentially expressed between *ptsd1*-YS and WT-YS (Appendix A). Among them, the expressions of the CK-degrading and CK-inactivating related homologs were highly expressed in *ptsd1* (Appendix A). Owing to the fact that CK-degrading and CK-inactivating related genes were highly expressed, the CK concentration in *ptsd1* was lower (Appendix A).

We identified 240 putative TF DEGs between *ptsd1*-YS and WT-YS. They belonged to 31 TF families (Appendix A). The WRKY (42) and NAC (32) families were the top two largest families having the most DEGs (Appendix A), and most of them were highly expressed in *ptsd1*-YS.

### 2.7. The Expression of Many Homeotic Genes Were Depressed in ptsd1

We isolated 28 differentially expressed MADS-box TFs from the four samples. The expressions of most MADS-box genes were reduced in *ptsd1*, such as the homologs of class E gene *sepallata/agamous-like gene 2* (*SEP/AGL2*), the class B genes *pistillata* (*PI*) and *apetala 3* (*AP3*) (Appendix A). The E-class MADS-box genes *TaSEP5* and *TaSEP6*, which specify organ identity and determine the spikelet meristem, were lowly expressed in *ptsd1*-YS during the young spike differentiation stages (Figure 4). The homeotic gene *TaFL*, which is associated with spikelet formation, was also lowly expressed in *ptsd1*-YS (Figure 4). Obviously, the expression of many homeotic genes was depressed in *ptsd1*.

### 2.8. DEGs Involved in Protein Modification and Calcium Signaling Were Highly Expressed in ptsd1-YS

Many DEGs involved in protein modification, such as receptor-like kinases, protein kinases, and phosphatases, were highly expressed in *ptsd1*-YS (Appendix A). Among the 307 protein modification related DEGs belonging to 40 subfamilies, 289 were highly expressed, and only 19 were lowly expressed in *ptsd1*. The receptor-like kinase (RLK/Pelle), legume lectin (L-LEC) subfamily (41) and DUF26-Ib subfamily (39) were the top three largest families having the most DEGs between *ptsd1*-YS and WT-YS (Appendix A). A total of 50 DEGs were related to Ca^2+^ signaling (Appendix A). Most Ca^2+^ signaling DEGs were highly expressed in *ptsd1*.

### 2.9. Ca^2+^ Influx and ROS Accumulation Caused Hypersensitive Response (HR)-Related PCD in ptsd1

The concentration of H_2_O_2_ in *ptsd1*-YS was significantly increased by 31% compared to WT-YS (Figure 5A). ROS is usually associated with plant defense response and HR. In our study, totally 140 DEGs, 131 highly and 9 lowly expressed in *ptsd1*-YS, were involved in ROS signaling (Appendix A), including genes encoding LRR receptor-like serine/threonine-protein kinase (FLS2), WRKY, mitogen-activated protein kinase (MAPK), cyclic nucleotide-gated ion channel (CNGCs), calcium-binding protein (CaMCML), calcium-dependent protein kinase (CDPK), respiratory burst oxidase homolog protein (Rboh), disease resistance protein (RPS2 and RPM1) and heat shock protein 90 (HSP90) (Appendix A). We proposed that the accumulation of cytosolic Ca^2+^, the activation of MAPKs, the ROS, and the expression of defense related WRKY TFs were involved in HR-related PCD, which was caused by oxidative damage in *ptsd1*. The ROS accumulation caused oxidative damage and cell death, which was consistent with the result of TUNEL assay that DNA degradation was widespread in the undifferentiated upper spikelets of the *ptsd1* (Figure 5G).

### 2.10. Degradation of Macromolecules in ptsd1

A distinct feature of *ptsd1* was the active massive degradation of macromolecules, such as proteins, polysaccharides, lipids, and nucleotides, as well as the biosynthesis of secondary metabolites. This was evidenced by the fact that, among the highly expressed genes of protein and amino acid, polysaccharide, lipid, nucleotide, and secondary metabolism in *ptsd1*, the ratios of the catabolism-related genes to anabolism-related genes were 12.75, 3.15, 1.14, 7.00 and 0.07 respectively (Appendix A). A total of 74 genes involved in potential ubiquitin-proteosome proteolytic pathways were highly expressed in *ptsd1* and this indicated that protein degradation via the 26S proteosome was active during the upper spikelet degradation processes in *ptsd1* (Appendix A). In addition to ubiquitin-proteosome proteolytic pathways, 25 genes encoding the proteinases were highly expressed in *ptsd1* (Appendix A). In *ptsd1*-YS, the contents of proteins were significantly decreased by 25.0%. (Figure 6A). The content of protein was lower in undifferentiated upper spikelets of *ptsd1* compared to the spikes of WT by histochemical observation (Appendix A). The content of total sugar was significantly decreased by 5.7% in *ptsd1* (Figure 6B). The chitinase encoding gene *TaCht*, which involved in degradation of chitin polysaccharide, was highly expressed in *ptsd1* (Figure 7A). The chitinase activity was 1.4-fold higher in *ptsd1* than that in WT (Figure 7B).

### 2.11. N recycling, Anthocyanin and Lignin Biosynthesis in ptsd1

In *ptsd1*, phenylalanine ammonia-lyase, guanine deaminase and fatty acid amide hydrolase encoding genes were highly expressed. N was released primarily from protein degradation, nucleic acid catabolism and amide hydrolysis (Appendix A, Figure 8A). The released N is assimilated, generally in the form of NH_4_^+^, via the glutamate decarboxylase and GS-catalyzed synthesis reaction and eventually produced glutamine (Figure 8A).

Metabolic indicators were measured to verify the hypothetical model of N recycling and anthocyanin and lignin biosynthesis gene regulatory network proposed according to the RNA-seq data in Figure 8. In *ptsd1*, the content of nitriate nitrogen was significantly decreased by 43.3%, while the content of NH_4_^+^ was significantly increased by 14.7% (Figure 6C,D). *TaGS1a* was highly expressed (Figure 2B), and the GS activity was significantly increased by 22.2% (Figure 6E). A total of seven ammonium transporter encoding genes, which participated in ammonium transport, were highly expressed in *ptsd1* (Figure 8A). Assimilation of NH_4_^+^ in plant cells is catalyzed by GS, which is important for nitrogen assimilation and recycling. Meanwhile, the content of lignin was significantly increased by 48.2% in *ptsd1* (Figure 6F), which was in agreement with the highly expressed lignin biosynthesis related genes (Figure 8B). Above all, the macromolecules, especially proteins, were degraded in *ptsd1*, the N was recycled in the form of NH_4_^+^, and the anthocyanin and lignin biosyntheses were enhanced in *ptsd1* (Figure 8).

The GST encoding genes were significantly highly expressed in *ptsd1* compared to WT (Appendix A, Figure 8A). The expression levels of *TaGSTF2* and *TaGSTU2* evaluated by qRT-PCR and RNA-seq were highly consistent (Figure 2B). *TaGSTF2* and *TaGSTU2* were highly expressed in *ptsd1* during the young spike differentiation stage from late double ridge stage to terminal spikelet stage (Figure 9A). The GST activity was 15.5-fold higher in *ptsd1*-YS compared to WT-YS at late double ridge stage, meanwhile, the GST activity in root, leaf and stem of *ptsd1* was also significantly high (Figure 9B). The glutathione-conjugate transporter genes, which might participate in the transport of glutathione-conjugate products, were highly expressed in *ptsd1* (Appendix A).

### 2.12. Co-Expression Clusters of the DE miRNAs

A total of 88 DE miRNAs were identified by the pairwise analysis (Appendix A) in the twelve samples (WT-st: S1a, S1b, S1c; WT-ys: S2a, S2b, S2c; *ptsd1*-st: S3a, S3b, S3c; *ptsd1*-ys: S4a, S4b, S4c, the samples WT-st, WT-ys, *ptsd1*-st and *ptsd1*-ys were prepared from the same original samples of WT-ST, WT-YS, *ptsd1*-ST and *ptsd1*-YS, respectively). The clean read number of each sample was more than 15.32 M, the average Q30 percentage was more than 98.42% (Appendix A).

The 88 DE miRNAs (Appendix A) were classified into four groups using *k*-means cluster. 23 miRNAs in cluster 1 were most highly expressed in *ptsd1*, 24 miRNAs in cluster 2 were most lowly expressed in *ptsd1*, twelve miRNAs in cluster 3 were most highly expressed in the young spikes and 29 miRNAs in cluster 4 were most highly expressed in the young stem tips (Appendix A). The miRNAs in cluster 1 and 2 were the key regulators causing to the phenotype of *ptsd1*. The qRT-PCR assay demonstrated that the expression patterns of the four representative miRNAs, novel-miR4, tae-miR159a, tae-miR167a and tae-miR9670-3p, were well consistent with that of the sequencing results (Appendix A).

### 2.13. DE miRNAs and Their Targets Involved in Senescence Related Biological Process in Mutant ptsd1

The transcriptome analysis indicated that senescence related biological processes were related to *ptsd1*, including senescence related signal response and senescence related metabolism response. Further analysis identified 22 key miRNAs associated with the senescence related biological processes (Appendix A). There were nine miRNA targets were involved in the hormone and Ca^2+^ signaling activation (Appendix A), ten receptor-like kinases in RLK/Pelle family and nine TFs were involved in the signal transduction and transcriptional regulation of senescence related metabolisms (Appendix A). Seventeen miRNA targets were involved in senescence related metabolisms (Appendix A). The tae-miR164 and novel-miR49 were extensively involved in senescence related biological processes, and they had nine and fourteen targets respectively (Appendix A). The novel-miR49 was lowly expressed in *ptsd1* compared to WT (Appendix A).

## 3. Discussion

### 3.1. The Undifferentiated Upper Spikelets of The Mutant Ptsd1 is A Kind of Senescence

At the cellular level, senescence can be classified into two categories, mitotic senescence and postmitotic senescence [29]. An example of mitotic senescence in plants is the arrest of shoot apical meristem, which is also called proliferative senescence [30]. In *Arabidopsis*, the arrest development of floral buds is related to an irreversible cessation of cell expansion and differentiation, as well as cell division [31]. Similarly, the upper spikelet primordia of *ptsd1* stopped differentiation at about the late double ridge stage, and the tissues absented the typical meristem characteristics, cells terminated division, differentiation, and expansion soon after (Appendix A). In this case, *ptsd1* should demonstrate a kind of mitotic senescence.

During wheat leaf senescence processes, proteins are degraded [32]. H_2_O_2_ plays an important role during the senescence process, and it could also be considered as a senescence signal and enhancer in different plant species, and it acts as an element of a complex regulatory network [33]. H_2_O_2_ accumulation and cell death indeed occurred in rice premature senescence leaf 85 (*psl85*) mutant [34]. The content of protein was lower in undifferentiated upper spikelets of *ptsd1* compared to that of WT, the oxidative damage and cell death caused by H_2_O_2_ accumulation were enhanced. These phenotypes of *ptsd1* were consistent with senescence.

### 3.2. Senescence Related Signal Response are Activated in ptsd1

#### 3.2.1. Hormone Signaling

Plant senescence is regulated by hormone homeostasis [35,36]. Plant CK levels are reduced during natural senescence, and CKs play a senescence-retarding role in plants [37]. The CK content of the transgenic *Arabidopsis* overexpression of CK oxidase/dehydrogenase (*AtCKX*) gene is reduced, which results in diminished activity of the vegetative and floral shoot apical meristems [38]. In our study, the CK content was reduced in *ptsd1*. Meanwhile, the expressions of nuclear localized *Arabidopsis* response regulator (ARR) like genes, which regulate the transcription of CK target genes [39], were lowly expressed in *ptsd1*. These data demonstrated that the CK signaling was repressed in *ptsd1*.

In addition to CKs, ET also plays a pivotal role in plant senescence. In our study, ET biosynthesis, response and signaling related genes were highly expressed in *ptsd1*. The 1-aminocyclopropane-1-carboxylic acid (ACC) synthase is a key enzyme in ET biosynthesis from the methionine, when expression of ACC synthase gene is enhanced, senescence of several plant species is activated [40]. The ACC synthase genes were highly expressed in *ptsd1* (Appendix A). Meanwhile, ET signaling related genes ethylene responsive 1 and *EIN3* were activated (Appendix A). *EIN3* is a *SAG* gene. The activity of EIN3 can be enhanced by ET that promotes leaf senescence [25].

#### 3.2.2. Ca^2+^ Signaling

The main symptoms of plant senescence are the dramatic increases in ROS [41]. The content of H_2_O_2_ was significantly increased in *ptsd1*-YS. Elevation of intracellular Ca^2+^ is essential for the H_2_O_2_ accumulation [42]. Normal cell function requires maintenance a low cytosolic Ca^2+^ concentration. Ca^2+^ concentration increases in the senescence processes [43] and Ca^2+^ influx is related to cell death [44]. In our study, Ca^2+^ signal transduction related genes were activated, such as cyclic nucleotide-gated ion channel protein genes which can mediate the initiation of the developmentally regulated cell death programs in a Ca^2+^ dependent manner [45], *CDPK* genes which participate in Ca^2+^ signal transduction, *Rboh* genes which can mediate the generation of superoxide through calcium-dependent NADPH oxidase were highly expressed in *ptsd1*.

#### 3.2.3. TFs

Many TFs play pivotal roles in plant senescence, such as the members of NAC and WRKY families [46]. These TFs have been functionally characterized in plant senescence, such as *AtORE1* [47], *AtNAP* [48], *TtNAM-B1* [49], *AtWRKY53* [50] and *AtWRKY70* [51]. Similarly, some WRKY and NAC TF genes were highly expressed in *ptsd1* (Appendix A). In contrast, the MADS-box floral homeotic genes were depressed in *ptsd1*. The E-class MADS-box genes *TaSEP5* and *TaSEP6* were the homologous genes of *OsMADS34* and *OsMADS5* in rice (Appendix A). Genetic and molecular analyses demonstrated that *OsMADS5* and *OsMADS34* together regulate floral meristem determinacy by positively regulating the other homeotic genes in rice [52]. In *ptsd1*, the functions of the homeotic genes need further research.

### 3.3. Senescence Related Metabolisms are Activated in ptsd1

The degradation of molecules such as proteins, lipids, nucleotides, and polysaccharides is a typical feature of senescence that occurs at the final phase of leaf development [53]. In our study, this process was reflected by the high expressions of several catabolism-related genes. Degradation of macromolecules, especially the proteins via the ubiquitin-proteasome system [54] was enhanced in *ptsd1*. Macromolecules were degraded and N was released in the form of NH_4_^+^ in *ptsd1*. Nitrogen is exported from senescing barley and wheat leaves as glutamate, but also as aspartate and threonine [18,55]. The relative abundance of glutamine increases during late senescence in wheat phloem [56]. The GS and asparagine synthetase genes, which participate in the N assimilation and remobilization [55] were highly expressed in *ptsd1*. Glutamine is the major mobile amino acid involved in the long-distance transport of N [57]. The degradation of macromolecules might be transported to other organs for recycling in the form of glutamine. *SAGs* are enriched in the biosynthesis of phenylpropanoids during the leaf senescence in sorghum (*Sorghum bicolor*) [58]. The phenylalanine ammonia-lyase genes, which encode the key enzyme in phenylpropanoid metabolic pathway, were highly expressed in *ptsd1*. The anthocyanin and lignin biosynthesis pathways were activated in *ptsd1* (Figure 8B), and the content of lignin was significantly increased in *ptsd1*.

All these data suggested that there were five key senescence related metabolisms were activated in *ptsd1* (Figure 8): (1) The proteins, lipids, polysaccharides and nucleotides were degraded; (2) The macromolecules degradation increased ammonium accumulation; (3) The ammonium was assimilated in the form of glutamine and aspartate; (4) The phenylalanine flowed into phenylpropanoid metabolic pathway as a result of some secondary metabolites biosynthesis, such as anthocyanin and lignin, and (5) GST catalyzed the reaction of glutathione with a wide variety of organic compounds to form R-S-glutathione, a process that was essential for the transport of a variety of metabolites.

### 3.4. The miRNAs Participate in Senescence Related Biological Process in ptsd1

Several studies have shown that microRNAs regulate plant senescence [59]. A conserved microRNA, miR164, is identified as a negative regulator of senescence [24]. ET enhances the activity of EIN3, EIN3 induces *NAC2* expression by directly repressing miR164 transcription, miR164 mediates the cleavage of *NAC2* and ultimately regulates the expression of a large set of *SAGs* through the feed-forward signaling cascade comprising *EIN3, miR164*, and *NAC* [24,25]. In this study, ET signaling related genes *EIN3* and TF genes in NAC family were activated, while tae-miR164 was lowly expressed in *ptsd1* (Appendix A). A novel microRNA, novel-miR49, was extensively involved in senescence related biological process by regulating 14 target genes (Appendix A). WRKY TFs play pivotal roles in leaf senescence [46], *WRKY* genes were regulated by seven microRNAs including novel-miR49 (Appendix A).

### 3.5. A Hypothesis of The Molecular Regulatory Network in ptsd1

In summary, we put forward a hypothesis of a molecular regulatory network in *ptsd1* (Figure 10): (1) The endogenous hormone ET is increased and CK is decreased, intracellular Ca^2+^ concentration is increased; (2) The abnormal phytohormones and Ca^2+^ signal are transduced through their respective signaling pathways; (3) The expressions of many *SAGs* are activated and some homeotic genes are depressed; (4) The meristem characteristics of the undifferentiated spikelet primordia are absent and senescence related metabolisms, including the degradation of macromolecules, N recycling, anthocyanin and lignin biosynthesis, and ROS accumulation, are activated.

Based on the research data of phenotype, metabolism, transcriptome and miRNome, we speculated that the activated senescence related process led to the upper spikelet primordia terminating differentiation in the end. However, the exquisite molecular genetic mechanism of the wheat spike differentiation requires further study.

## 4. Materials and Methods

### 4.1. Plant Materials and Growth Conditions

The wheat cultivar “Guomai 301” was bred in our laboratory. The seeds of WT were treated with EMS and planted in Shangqiu Experimental Farm, Henan Province, China (34°25′ N, 115°39′ E, 49 m asl) in 2012–2015. The mutant *ptsd1* was obtained at M_2_ generation in 2014–2015 growing season. Mutant lines of *ptsd1* had truncated spikes without segregating at M_3_ generation in 2015–2016 growing season, which primarily indicated that the phenotype was controlled by recessive genes. The field experiments were carried out in a completely randomized design as described by Duan et al. [60].

### 4.2. Morphological Observation and Analysis

Tiller number, spike number, plant height, internode number of the main stem and etc. of *ptsd1* and WT were observed and measured. Each genotype sample was prepared by random taking ten individuals. The young spikes were observed as described by He et al. [61]. For observation with scanning electron microscope, young spikes were fixed in 1% osmic acid solution prepared in 0.1 mol/L phosphate buffer at pH 7.4 for one hour, dehydrated in a graded ethyl alcohol series for fifteen minutes in each solution, and critical point dried using CO_2_. The dried tissues coated with platinum were observed and photographed using a scanning electron microscope (SU8010, Hitachi, Tokyo, Japan).

### 4.3. RNA Extraction, mRNA and miRNA Sequencing

The young spikes at late double ridge stage [28] and young stem tips (short stem fragment just connected with young spikes) of WT and mutant *ptsd1* were identified and sampled for RNA extraction [62]. Total RNAs of WT and mutant *ptsd1* plants were extracted with TRIzol^®^ reagent (TransGen Biotech, Beijing, China). All the samples for mRNA and miRNA sequencing had three biological replicates. RNA sequencing and basic analysis were carried out in BioMarker Company (Beijing, China). Sequences have been deposited at the Sequence Read Archive of the National Center for Biotechnology under BioProject numbers PRJNA553833 and PRJNA554945.

### 4.4. mRNA Data Analysis

The clean reads were mapped to the reference wheat genome (ftp://ftp.ensemblgenomes.org/pub/plants/release-32/fasta/triticum_aestivum; access date: 8 September 2016) using Tophat2 tool software [63]. Gene functional annotation was carried out as described by He et al. [61]. Gene expression levels were estimated by FPKM [64]. DEGs between two sample groups were analyzed using the DESeq R package [65]. The FDR < 0.01 (false discovery rate) and FC ≥ 2 (fold change) were set as the thresholds for significantly DEGs.

Pearson’s correlation coefficient between biological replicates was calculated using the RPKM values [66]. The *k*-means cluster was performed on the BMKCloud platform (https://www.biocloud.net/) with DEGs (FDR < 0.01 and FC ≥ 4). For the single gene, heat maps were drawn using the software HemI (http://hemi.biocuckoo.org; access date: 5 November 2014) according to the FPKM values. For the overall trend of the DEGs in the families, heat maps were drawn according to the average FPKM values in a family.

### 4.5. miRNA Data Analysis

The clean reads were used to identify known miRNAs and predict novel miRNAs by comparing with known miRNAs from miRBase 21 (http://www.mirbase.org; access date: 3 July 2014). The miRNA expression levels were defined as “counts of reads mapped to miRNA × 1,000,000”/“reads mapped to the reference genome” (TPM) [67]. Differentially expressed miRNAs (FDR ≤ 0.05 and FC ≥ 2) between samples were screened out using DESeq software (BioMarker, Beijing, China) [68].

The miRNA target gene prediction was performed using TargetFinder [69]. Briefly, miRNA sequences matched to the reference mRNA sequences and potential targets were computationally predicted by the match/mismatch-scoring ratio. Mismatched pairs were scored as 1, and G:U pairs were scored as 0.5. Only predicted targets with scores less than four were considered reasonable.

### 4.6. qRT-PCRs of mRNAs and miRNAs

The qRT-PCR was performed as described by He et al. [61]. For miRNA, reverse transcription was performed with 1 μg RNA using the miScript reverse transcriptase mix (Qiagen, Beijing, China) according to the manufacturer’s protocol. The miRNA specific forward primer is the same as the mature miRNA sequence. *U6* was used as an internal control. Real-time PCRs of miRNAs were performed using the miScript SYBR Green PCR kit (Qiagen, Beijing, China) with the specific primers and the universal primer provided in the kit following the production instructions. All primer sequences are listed in Appendix A.

### 4.7. Histochemical Observations

The young spikes of WT and mutant *ptsd1* were fixed in FAA solution (5 mL of formalin, 5 mL of acetic acid and 90 mL of 70% ethyl alcohol). The samples were dehydrated, embedded in paraffin and sectioned with a rotary microtome as described by Geng et al. [70]. The cell morphology was observed after the tissues were stained with saffron and solid green (G1031, Servicebio, Wuhan, China). The distributions of starch and protein were observed after the tissues were stained with Periodic Acid-Schiff and Naphthol Yellow S (G1068, Servicebio, Wuhan, China). Photos were taken with a camera (Nikon Eclipse E100, Tokyo, Japan) and analyzed with CaseViewer software.

### 4.8. TUNEL Assays

The sections of the young spikes of WT and mutant *ptsd1* at late double ridge stage were prepared as described by Phan et al. [71]. The terminal deoxynucleotidyl transferase-mediated dUTP-biotin nick end labeling (TUNEL) assay was performed using a kit (Fluorescein In Situ Cell Death Detection Kit; Roche, Indianapolis, IN, USA) according to the manufacturer’s instructions. The green fluorescence of fluorescein and blue fluorescence of 4′,6-diamidino-2-phenylindole (DAPI) in the tissues were activated and observed under a fluorescence scanning confocal microscope (Nikon Eclipse Ti-SR).

### 4.9. Determination of Metabolic Indicators

The young spikes of WT and mutant *ptsd1* at late double ridge stage were dissected out, frozen immediately in liquid nitrogen and stored at −80 °C. The contents of ZA, total protein, total sugar, nitrate nitrogen and ammonium nitrogen (NH^4+^), lignin and H_2_O_2_ and the activities of glutathione *S*-transferase, glutamine synthetase and chitinase were measured. Each measurement was repeated for three times. The phenotyping of the 10 physiological traits were performed using different assay kits according to the manufacturer’s instructions in Comin Biotechnology Co., Ltd. (Suzhou, China). The detail calculation formulas and results were described in Appendix A.

#### 4.9.1. Determination of ZA Content

The ZA of young spikes was extracted from 100 mg freeze-dried meal in 1 mL mixed solution (carbinol: 0.5% acetic acid = 80:20) at 4 °C overnight, centrifuged at 8000× *g* for 10 min. The sample was decolored by adding 0.5 mL petroleum ether for three times, discarded the supernatant petroleum ether. The pH value was adjusted to 8, then extracted with acetic ether for three times, combined the supernatant organic phases and blew dry with a nitrogen blower at 60 °C. The sample was ready to determine by adding 0.5 mL mobile phase solution and filtered with a needle filter. ZA was determined by HPLC (Agilent 1100) system with a Kromasil C18 (250 mm × 4.6 mm, 5 μm) reversion phase chromatography column at 35 °C. The mobile phase composed of carbinol and 1% acetic acid solution (4:6, *v*/*v*) at a flow rate of 0.8 mL/min, and the UV absorbance was monitored at 254 nm. The chromatography elution profiles of the standard ZA and a sample were shown in Appendix A.

#### 4.9.2. Determination of Soluble Protein and Total Sugar Contents

The soluble proteins of WT and mutant *ptsd1* were extracted using 100 mg young spikes in a 50 mM cold potassium-phosphate solution at pH 7.0, centrifuged at 10,000× *g* at 4 °C for 15 min. The supernatant was used to quantify the soluble protein. The soluble protein content was determined with the colorimetric assay described by Smith et al. [72].

The sugars of each sample were extracted from 100 mg freeze-dried meal. 1 mL of 6 M HCl was added to each sample, heated in water bath at 95 °C for 30 min, then added 1 mL of 100 g/L NaOH, mixed by vortex, centrifuged at 8000× *g* for 10 min. The supernatant was used for quantification of the total sugars. The sugars were quantified using 3,5-dinitrosalicylic acid method described by Breuil and Saddler [73].

#### 4.9.3. Determination of Nitrate Nitrogen and Ammonium Nitrogen Content

The nitrate nitrogen and ammonium nitrogen of young spikes were extracted from 100 mg freeze-dried meal in 1 mL of distilled water at 90 °C for 30 min. The nitrate nitrogen content of the solution was quantified spectrophotometrically after mixing 0.02 mL of solution with 10% (*w*/*v*) salicylic acid in 96% sulfuric acid [74]. The ammonium content was determined with the colorimetric assay described by Krom [75].

#### 4.9.4. Determination of H_2_O_2_ Content

The H_2_O_2_ was extracted using 100 mg young spikes in the dark in 3 mL of 25 mM phosphate solution, pH 7.0, containing 0.05% guaiacol and 2.5 units horseradish peroxidase at 25 °C for 2 h. Absorbance of the solution was measured at 450 nm as described by Tiedemann [76].

#### 4.9.5. Determination of Lignin Content

The young spikes of WT and mutant *ptsd1* were dried in an air-forced oven at 80 °C. Two milligrams of the dried tissue were ground into fine powder and homogenized in 0.5 mL of 25% (*v*/*v*) acetyl bromide in glacial acetic acid at 80 °C for 40 min. And then the extraction solution was mixed with 0.5 mL of acetic acid. The supernatant was used for quantification of total lignin. Analysis of lignin content was carried out according to the method described by Li et al. [77].

#### 4.9.6. Determination of GST Activity

The young spikes of WT and mutant *ptsd1* were homogenized in 1 mL of 0.1 M sodium phosphate solution, and centrifuged at 8000× *g* for 10 min. The supernatant was used for determination of GST activity. A total of 220 μL solution (supernatant, 20 μL; 1-chloro-2,4-nitrobenzene solution, 180 μL; glutathione solution 20 μL) was prepared for determination of GST activity, and then the absorbance was determined and recorded as A1, A2 at the time point 10 s, 310 s respectively with a spectrometer at 340 nm. Detailed calculation formulas and results are described in Appendix A.

#### 4.9.7. Determination of Total GS Activity

One-hundred milligrams frozen young spikes were homogenized in 1 mL of extraction solution containing 50 mM Tris-HCl (pH 8.0), 2 mM MgSO_4_, 0.4 M sucrose, 2 mM dithiothreitol. The homogenate was centrifuged at 8000× *g* at 4 °C for 10 min. The supernatant was used for determination of GS activity. The GS activity was determined according to the modified method of Yu et al. [78].

#### 4.9.8. Determination of Chitinase Activity

One-hundred milligrams frozen young spikes were homogenized in 1 mL of 10 mM sodium acetate solution (pH 5.0) and centrifuged at 10,000× *g* for 20 min. The supernatant was used for the determination of chitinase activity. The enzymatic activity was determined spectrophotometrically by measuring the reaction products from the chitinase-catalyzed hydrolysis of chitin at 585 nm as described by Mohammadi et al. [79].

## Figures and Tables

**Figure 1 ijms-20-04642-f001:**
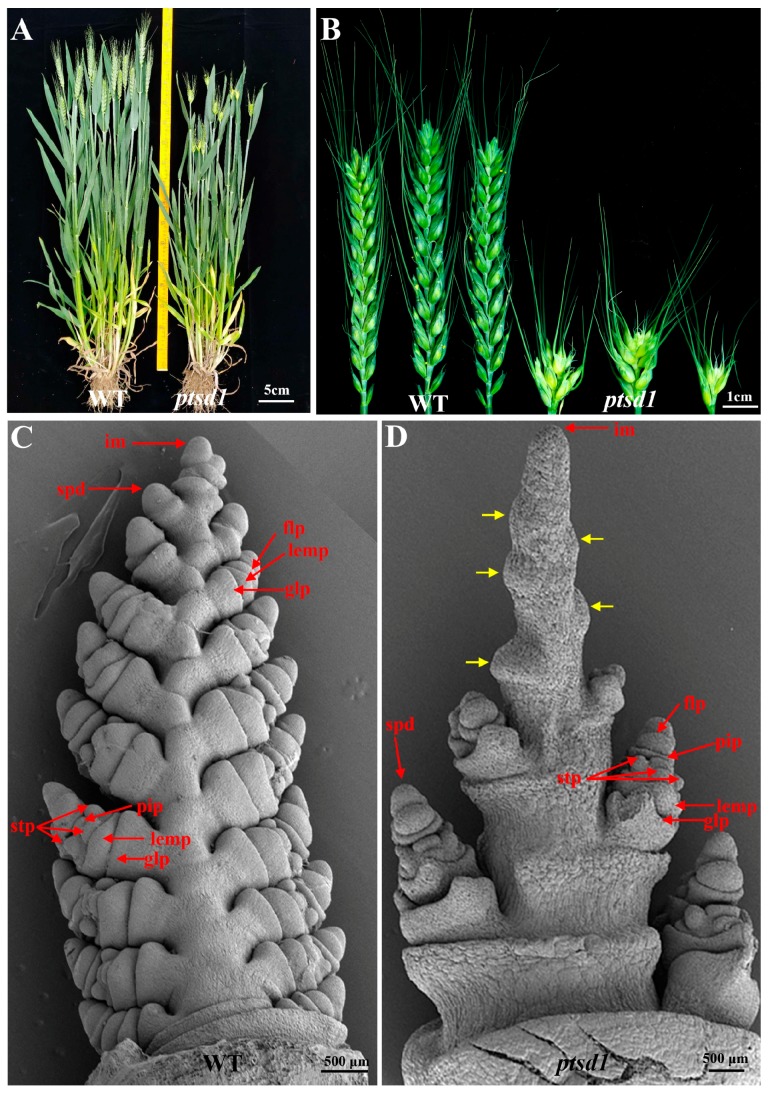
The individual plants and spikes of wild type (WT) and mutant *ptsd1*. (**A**) The plant phenotype of WT (left) and mutant *ptsd1* (right); (**B**) The spikes of WT (left) and mutant *ptsd1* (right); (**C**) The ultrastructure of the young spikes on WT at terminal spikelet stage; (**D**) The ultrastructure of the young spikes on mutant *ptsd1* at late terminal spikelet stage. Yellow arrowheads indicate the undifferentiated upper spikelet primordia. im, inflorescence meristem; spd, spikelet meristematic dome; glp, glume primordium; lemp, lemma primordium; flp, floret primordium; stp, stamen primordium; pip, pistil primordium.

**Figure 2 ijms-20-04642-f002:**
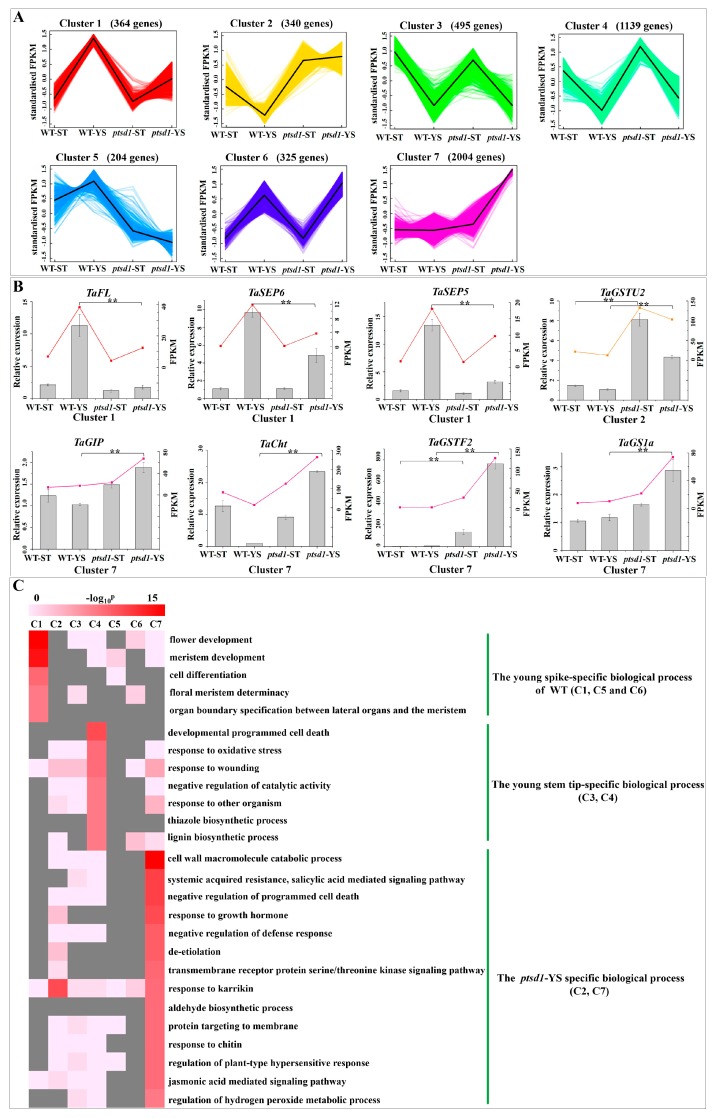
The expression profiles and functional enrichment of the differentially expressed genes (DEGs). (**A**) *K*-means clusters of the DEGs. The X-axis indicates the samples, Y-axis indicates the normalized fold-changes of the gene expression levels. (**B**) The expression patterns of some key genes verified by real-time qRT-PCR. The X-axis indicates samples. The left Y-axis indicates relative gene expression levels determined by qRT-PCR and the expression values were adjusted by setting the minimum expression as 1 for each gene. The right Y-axis indicates the fragments per kilobase of transcript per million fragments mapped (FPKM). The asterisk indicates the significant difference between different samples (**: *p* ≤ 0.01). (**C**) GO-term function enrichment of different clusters. The significances of the most represented GO-slims in each main cluster are indicated using log-transformed *p*-value (red). The dark grey areas represent the missing values. C1–C7: cluster1–cluster7.

**Figure 3 ijms-20-04642-f003:**
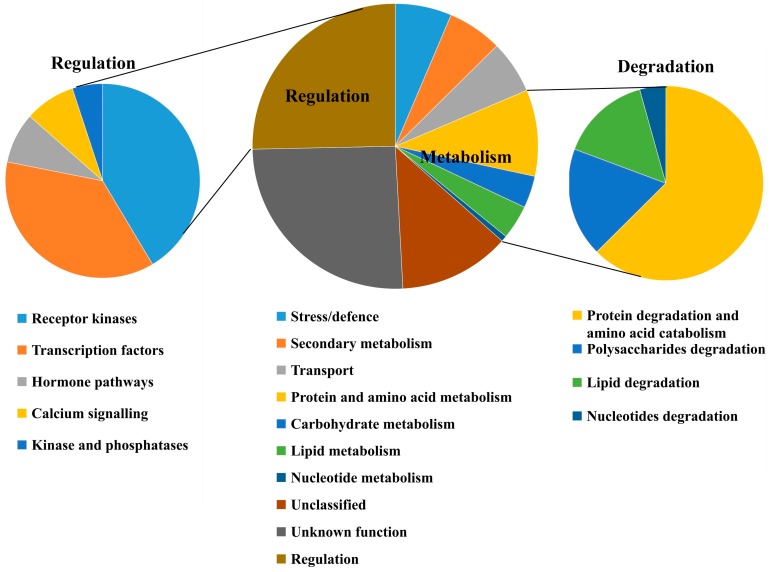
Potential functions of the DEGs between *ptsd1*-YS and WT-YS. The groups are illustrated by the central pie chart with extra charts. Left chart indicates the genes encoding putative regulatory proteins. Right chart indicates the genes involved in macromolecule degradation of four major nutrients. The gene annotation information is listed in Appendix A.

**Figure 4 ijms-20-04642-f004:**
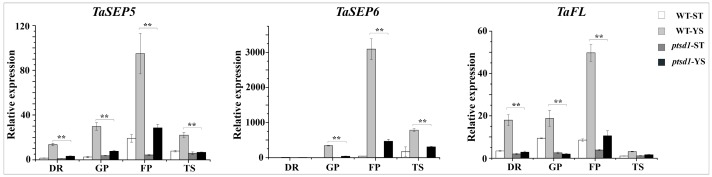
Expression patterns of three homeotic genes. Double ridge stage (DR), glume primordium visible stage (GP), floret primordium visible stage (FP) and terminal spikelet stage (TS). The asterisk indicates the significant difference between different samples (**: *p* ≤ 0.01).

**Figure 5 ijms-20-04642-f005:**
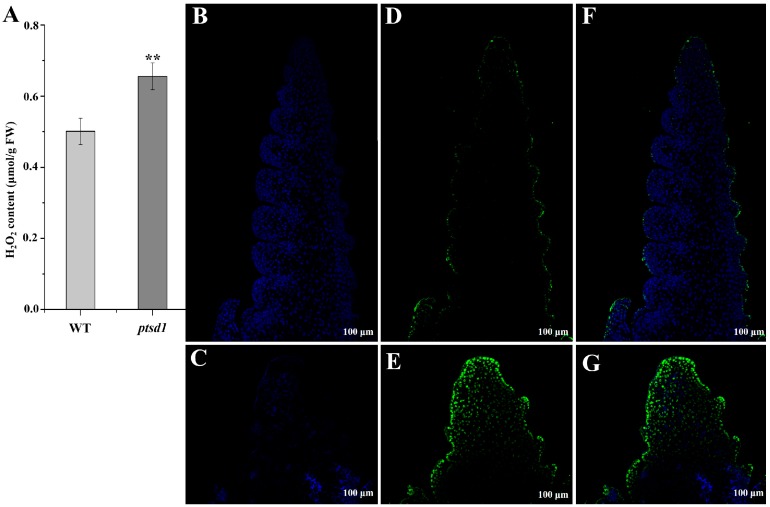
The H_2_O_2_ contents (**A**) and TUNEL assay (**B**–**G**) of WT and *ptsd1*. (**A**) The H_2_O_2_ contents in young spikes of WT and *ptsd1*. The asterisks represent the significant difference between WT and *ptsd1* (**: *p* ≤ 0.01). FW, fresh weight. (**B**–**G**) TUNEL assay results of WT and *ptsd1*. The young spikes were at late double ridge stage. The spikes of WT (**B**) and *ptsd1* (**C**) stained with DAPI. Positive stained results of WT (**D**) and *ptsd1* (**E**) with fluorescein. (**F**) is the overlap picture of (**B**,**D**); (**G**) is the overlap picture of (**C**,**E**). Green indicates the programmed cell death (PCD).

**Figure 6 ijms-20-04642-f006:**
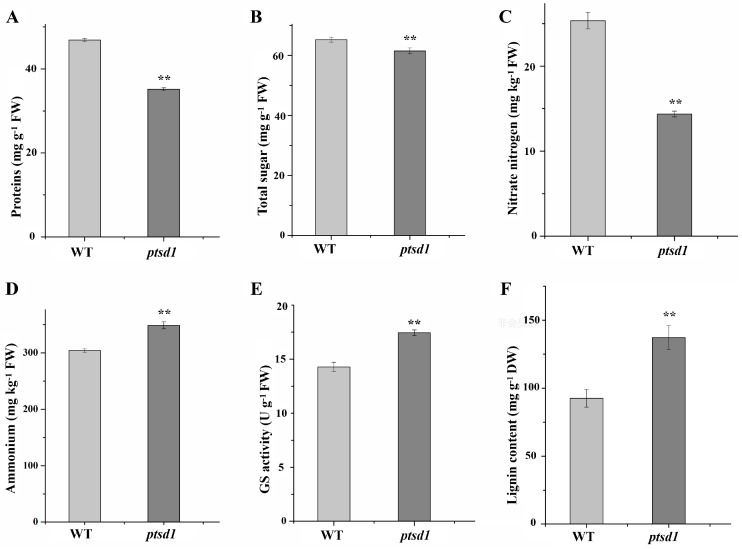
Physiological traits of WT and *ptsd1*. The contents of proteins (**A**), total sugar (**B**), nitrate nitrogen (**C**), ammonium nitrogen (**D**), GS activity (**E**) and lignin (**F**) in spikes of WT and *ptsd1*. Each sample has three independent biological replicates. Bars indicate the standard deviation. FW, fresh weight; DW, dry weight. The asterisks represent the significant difference between WT and *ptsd1* (**: *p* ≤ 0.01).

**Figure 7 ijms-20-04642-f007:**
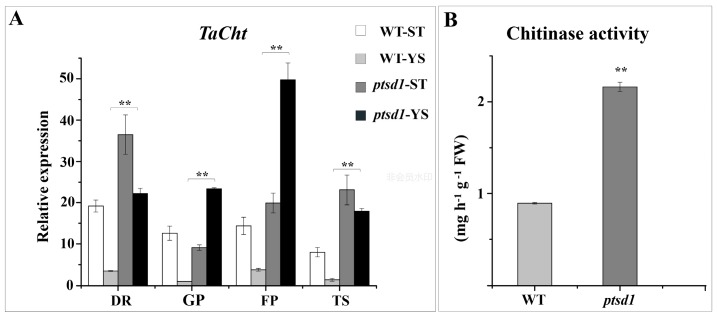
Expression pattern of chitinase gene *TaCht* and chitinase activity in spikes of WT and *ptsd1*. The asterisks represent the significant difference between WT and *ptsd1* (**: *p* ≤ 0.01).

**Figure 8 ijms-20-04642-f008:**
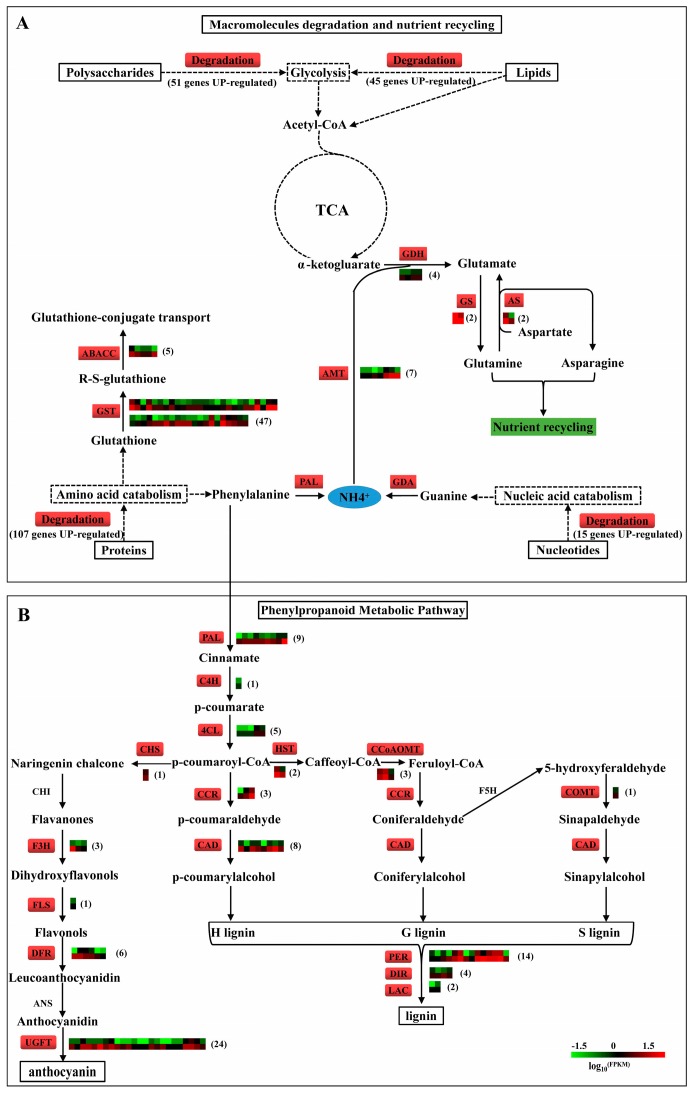
The key metabolic pathway and regulatory network in mutant *ptsd1*. (**A**) Macromolecule degradation and nutrient recycling in mutant *ptsd1*. (**B**) Overview of putative phenylpropanoid metabolic pathway involved in anthocyanin and lignin biosynthesis in *ptsd1*. TCA, tricarboxylic acid cycle; the box includes the critical enzymes comprising the entry pathway. Enzymes coloured in red indicate the highly expressed genes in *ptsd1*-YS, the heatmaps show the expression profiles between WT-YS (upside) and *ptsd1*-YS (downside), the figures in the parentheses are the numbers of DEGs. The real lines indicate the direct products, the dotted lines indicate the indirect products and metabolic process. ABCC3, ABC transporter C; GST, glutathione S-transferase; AMT, ammonium transporter; GDA, guanine deaminase; GDH, glutamate decarboxylase; GS, glutamine synthetase; AS, asparagine synthetase; PAL, phenylalanine ammonia-lyase; C4H, cinnamate 4-monooxygenase; 4CL, 4-coumarate-CoA ligase; CHS, chalcone synthase; CHI, chalcone isomerase; F3H, naringenin, 2-oxoglutarate 3-dioxygenase; FLS, flavonol synthase; DFR, dihydroflavonol-4-reductase; ANS, anthocyanidin synthase; UFGT, anthocyanidin 5,3-*O*-glucosyltransferase; HST, shikimate O-hydroxycinnamoyl- transferase; CCR, cinnamoyl CoA reductase; CAD, cinnamyl alcohol dehydrogenase; CCoAOMT, caffeoyl-CoAO-methyltransferase; COMT, tricetin 3,4,5apos-*O*- trimethyltransferase; PER, peroxidase; DIR, dirigent protein; LAC, laccase. The detailed information of the DEGs is listed in Appendix A.

**Figure 9 ijms-20-04642-f009:**
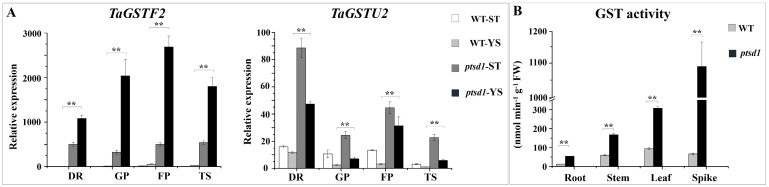
Expression patterns of GST encoding genes and GST activity in WT and *ptsd1*. The asterisk indicates the significant difference between different samples (**: *p* ≤ 0.01).

**Figure 10 ijms-20-04642-f010:**
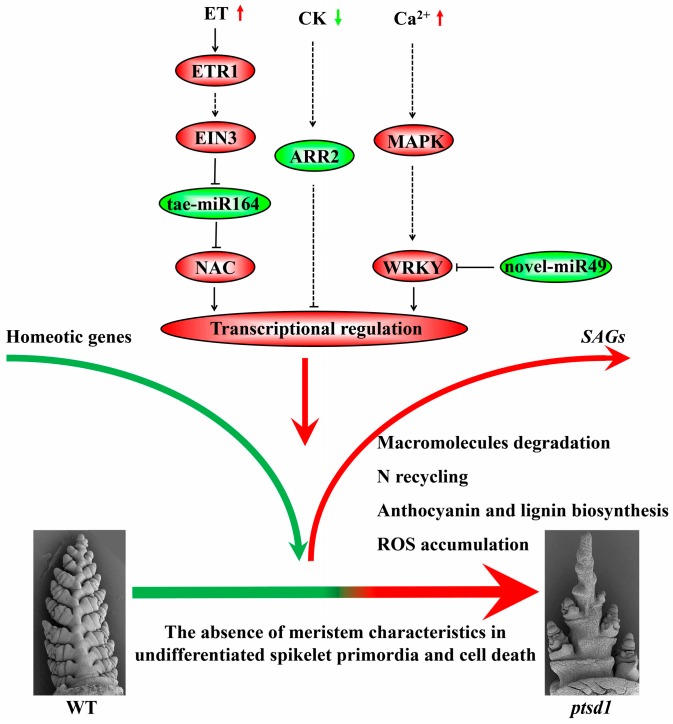
A hypothesis of molecular regulatory network in *ptsd1*. The modules coloured in red are highly expressed in *ptsd1*-YS and modules coloured in green are lowly expressed in *ptsd1*-YS. The real lines indicate the direct interaction, the dotted lines indicate the indirect interaction.

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
