# Peer review of "Enhanced Senescence Process is the Major Factor Stopping Spike Differentiation of Wheat Mutant ptsd1"

_ijms, 2019, doi:10.3390/ijms20184642_

Round 1

Reviewer 1 Report

The work is very interesting. Authors have very accurately described both results and material s and methods. Discussion are very substanzial. The decision to divide them into  several paragraphs was a good. Single discussion would have required more time to compare the results of individual experiments with the discussion. For me the work can be published in the present form.

Author Response

Revision list according to the comments from Reviewer 1

Point 1: The work is very interesting. Authors have very accurately described both results and materials and methods. Discussion are very substantial. The decision to divide them into several paragraphs was a good. Single discussion would have required more time to compare the results of individual experiments with the discussion. For me the work can be published in the present form.

Response 1: Thank you very much for your positive comments to our work.

Reviewer 2 Report

The paper "Enhanced Senescence Process is the Major Factor Stopping Spike Differentiation of Wheat Mutant ptsd1" by Jiao et al.  provides characterization of a wheat spike mutant ptsd1 in terms of transcriptomics, metabolites and morphology. The work i quite comprehensive and provides valuable data.

I think it is a weakness that the authors apparently failed to backcross the mutant to the parent cultivar Guomai 301. Subsequent segregation analysis would have provided valuable insight regarding the nature of the mutation (is it a single gene? is it dominant or recessive?). Segregating plants could also have been used to confirm that observations at the molecular level correlate with the phenotype. If the authors did backcross, I think they should include that data.

I think the authors should discuss the microRNA "novel-miR49" further. It should be clear from the main text if this microRNA was found in wild type and mutant and weather it was differentially expressed.

The manuscript is generally well writen but too informal in places ("whats more" "didn't")

Author Response

Revision list according to the comments from Reviewer 2

Point 1: I think it is a weakness that the authors apparently failed to backcross the mutant to the parent cultivar Guomai 301. Subsequent segregation analysis would have provided valuable insight regarding the nature of the mutation (is it a single gene? is it dominant or recessive?). Segregating plants could also have been used to confirm that observations at the molecular level correlate with the phenotype. If the authors did backcross, I think they should include that data.

Response 1: Actually, we have backcrossed the mutant to the parent cultivar Guomai 301, and we also have made crosses of ‘CS × ptsd1’ and ‘Zhoumai18 × ptsd1’. The spike phenotype of F1 was normal. Subsequent segregation analysis indicates that it’s regulated by recessive genes according to the segregation ratio of F2 populations. The primary result has been given in the ‘introduction’ section.

This manuscript is to report the molecular regulatory network of mutant ptsd1 for the abnormal spike development. The results of the heredity, mapping and cloning of the mutated gene ptsd1 obtained by forward genetic studies will be organized into another manuscript as soon as possible. So the heredity data haven’t placed in this manuscript in detail.

Point 2: I think the authors should discuss the microRNA "novel-miR49" further. It should be clear from the main text if this microRNA was found in wild type and mutant and weather it was differentially expressed.

Response 2: According to your suggestion, we have added more detailed data on the differentially expressed pattern of the microRNA "novel-miR49" in the main text to ensure the readers can fully understand our work. 

Point 3: The manuscript is generally well writen but too informal in places ("whats more" "didn't")

Response 3: We have checked the whole article carefully and deleted the informal expression ‘whats more’ in the text line 217, 271 and 376.

Reviewer 3 Report

The manuscript deals with huge work. The data are sound and support fully the conclusion. Results are presented clearly and effectively, and the multiple experiments are very well designed.

I recommend accepting the manuscript as it is, and I do my congratulations to the authors.

Author Response

Revision list according to the comments from Reviewer 3

Point 1: The manuscript deals with huge work. The data are sound and support fully the conclusion. Results are presented clearly and effectively, and the multiple experiments are very well designed. I recommend accepting the manuscript as it is, and I do my congratulations to the authors.

Response 1: Thank you very much for your positive comments to our work.